# A Low-Cost Hardware Architecture for EV Battery Cell Characterization Using an IoT-Based Platform

**DOI:** 10.3390/s23020816

**Published:** 2023-01-10

**Authors:** Rafael Martínez-Sánchez, Ángel Molina-García, Alfonso P. Ramallo-González, Juan Sánchez-Valverde, Benito Úbeda-Miñarro

**Affiliations:** 1Department of Automatics, Electrical Engineering and Electronic Technology, Universidad Politécnica de Cartagena, 30202 Cartagena, Spain; 2Department of Information and Communications Engineering, Computer Science Faculty, Universidad de Murcia, 30100 Murcia, Spain

**Keywords:** battery regeneration, NIMH characterization, battery cell diagnosis, low-cost solution

## Abstract

Since 1997, when the first hybrid vehicle was launched on the market, until today, the number of NIMH batteries that have been discarded due to their obsolescence has not stopped increasing, with an even faster rate more recently due to the progressive disappearance of thermal vehicles on the market. The battery technologies used are mostly NIMH for hybrid vehicles and Li ion for pure electric vehicles, making recycling difficult due to the hazardous materials they contain. For this reason, and with the aim of extending the life of the batteries, even including a second life within electric vehicle applications, this paper describes and evaluates a low-cost system to characterize individual cells of commercial electric vehicle batteries by identifying such abnormally performing cells that are out of use, minimizing regeneration costs in a more sustainable manner. A platform based on the IoT technology is developed, allowing the automation of charging and discharging cycles of each independent cell according to some parameters given by the user, and monitoring the real-time data of such battery cells. A case study based on a commercial Toyota Prius battery is also included in the paper. The results show the suitability of the proposed solution as an alternative way to characterize individual cells for subsequent electric vehicle applications, decreasing operating costs and providing an autonomous, flexible, and reliable system.

## 1. Introduction

Different international policies are promoting renewables and strategies to decrease fossil-fuel dependence to reduce their emissions. For example, highly concerned about climate change, EU countries are promoting agreements to cut their emissions by 80% by 2050, with the aim of restricting global warming to under 2 °C [1]. These efforts are shared among most sectors, with the road transport sectors expected to reduce their emissions by 95% [2]. Moreover, they highly depend on oil, which also raises concerns related to supply security and resource depletion [3]. Subsequently, it can be affirmed that internal combustion engine (ICE)-based vehicles, industries, and factories cause remarkable global warming and environmental pollution because of enormous carbon gas emissions. For this reason, recent contributions are mostly focused on proposing electric vehicle (EV)-based solutions, which are becoming more attractive to different sectors and researchers [4,5] because such EVs use electrical energy to efficiently operate motors [6]. In addition, EVs can become an important alternative for sustainable development in developing countries, such as BRICS members (Brazil, Russia, India, China, and South Africa). In fact, Martins et al. [7] affirmed that EVs could bring environmental benefits and effective strategies in the long term if policies encouraging their use are promoted in line with fleet development.

By considering the battery cost reduction during the last decade, Haram et al. [8] estimated that the battery demand could be increased 14-fold by 2030 compared to 2018. Indeed, EV applications would dominate the battery demand by over 88% in comparison to other applications, such as consumer electronics or energy storage systems (ESS). Nevertheless, new concerns are currently being raised regarding the expected quantity of retired EV batteries (REVBs) exposed to the environment when they reach 70–80% of their original capacity [9]. In fact, the growing environmental concerns related to discarded EV batteries have led engineers and policymakers to consider ESS solutions as an application to utilize used EV batteries. Subsequently, the concept of repurposing EV batteries for different applications has become a prominent solution. In addition, new battery design strategies, recycling routes, and reverse logistics have recently been proposed to overcome environmental concerns and future perspectives based on alternative policies for sustainable development [10]. Recent pilot projects and studies analyzed the lifetime extension of EV batteries by using them in other applications, leading to remarkable benefits, such as economic, environmental, and social benefits. However, due to both the limited availability of data and the novelty of the topic, Bobba et al. [11] recently affirmed that more investigations are needed to quantify and confirm such benefits.

From the specific literature, different contributions can be found regarding second-life EV batteries depending on their cell types and degradation rates. Moreover, recent studies have also characterized battery performance in accordance with energy efficiency, discharge capacity, or thermal response. White et al. [12] modeled potential second-life lithium iron phosphate (LFP) modules by considering their discharge capacity retention with a state of health (SOH) near 80%. Results gave energy efficiency values higher than 95% when discharge periods were slower than a 1 h rate. Additional experimental results can be found in [13], showing that LFP cells outperformed nickel cobalt aluminum (NCA) cells under peak-shaving duty cycles and second-life frequency regulation in terms of lifetime capacity loss and energy efficiency. Martínez-Laserna et al. [14] compared high-power intermittent renewable smoothing and low-power residential demand response to second-life nickel manganese cobalt (NMC) cell degradation. The results showed the particularly detrimental effect of internal resistance growth and the “aging knee” for high-power second-life applications. Different second-life Nissan Leaf battery modules with lithium manganese oxide (LMO) were tested, beginning near a 65% SOH under 1 h of accelerated degradation testing in [15]. From these results, a useful second life of 5 years was estimated in solar firming applications. Elliot et al. [16] analyzed LFP and NMC-LMO cells, cycled under frequency regulation duty cycles and energy arbitrage. They concluded that NMC-LMO degraded at twice the rate of LFP, causing energy arbitrage twice the frequency regulation degradation rate. Nevertheless, researchers agree that a relevant drawback for potential second-life battery considerations is related to the diversity and heterogeneity of EV battery models, in terms of shape, size, chemistry, cell format, and thermal management [17]. In fact, this heterogeneity leads to remarkable differences in their characteristics, such as energy efficiency, energy capacity, thermal response, or degradation rates. Subsequently, recent studies established that additional comparative experimental tests are needed to characterize such different EV batteries in detail [18]. Variations in terms of ramp rate, power range, cycling frequency, or discharge and charge duration should be analyzed in more detail before proposing their use for specific grid services. Moreover, a suitable selection of second-life energy storage applications for EV batteries should be carried out on the basis of these comparative experimental studies. According to the specific literature, there are several examples based on a real-time Battery Monitoring System (BMS) [19,20], as well as electrochemical–thermal battery models [21], or battery health estimations [22]. Most of these solutions are based on Internet of Things (IoT) technology, which concerns the interconnection of physical objects through the Internet, in order to share information and coordinate decisions based on sensed data [23]. In fact, Eris et al. [24] concluded that IoT enables real-time vehicle performance monitoring. BMS solutions have also been utilized to indicate the battery state-of-charge (SOC) [25]. These monitored parameters allow the control of actions to maintain the battery’s lifecycle and safety against potential hazards. However, there is a lack of contributions focused on characterizing and automatically evaluating individual EV cells under the potential abnormal performance of any specific cell [26]. Moreover, Ghuge et al. [27] affirmed that BMS was unable to sense movable connections present in the battery, corrosion of connections, or leakage of cell material. By considering the recent literature, this paper proposes and assesses a low-cost hardware architecture for EV battery cell characterization using an IoT-based platform. This solution allows characterizing the individual cells of such batteries, identifying potentially abnormal cell performance, and providing an alternative way to recover batteries for second-life applications in electric vehicles, minimizing costs. The methodology is based on diagnosing and proposing a battery regeneration solution with an alternative individual analysis of each cell of the battery. The proposed system performs a discharge to each individual cell, in a controlled and automated manner, by using a constant electric current until reaching a minimum cell voltage value. As a novelty, in comparison with other approaches, this solution allows the identification of such individual cells with potential damages or abnormal performance, allowing the reuse of the rest of the cells and regenerating in a more sustainable manner. The proposed system is based on low-cost hardware and nonproprietary platforms with the aim of giving a flexible, open, and highly configurable solution in an IoT environment.

The remainder of the paper is structured as follows: Section 2 discusses the materials and methods for this specific application, considering both hardware and software developments; Section 3 describes the case study based on a commercial Toyota Prius battery; results are presented in Section 4; and lastly, Section 5 gives the conclusions.

## 2. Materials and Methods

For the construction of the system, aimed to diagnose each cell and initiate the charging and discharging processes autonomously based on IoT, both the architecture and the hardware and software of the system were designed. The system is able to perform several functions:The state of each individual cell is independently diagnosed, according to the amount of electrical energy discharged in each cycle (Wh).Both charging and discharging processes of each cell are automatically carried out in a consecutive manner, and under constant current conditions. The system can measure the voltage (V) of the cells to establish when the voltage has reached minimum discharge and maximum charge voltage on each cycle. The charge and discharge current (mA) are constantly monitored in order to automatically correct this parameter with the aim of producing charges and discharges at constant currents.The values of voltage (V), current (mA), energy (Wh), and temperature (°C) on each cell are continuously measured and stored independently for each charge and discharge period, described in the previous stage.The process finishes if the corresponding established threshold or safety values were reached: maximum charging voltage (V), minimum discharging voltage (V), and maximum temperature of the cell (°C).Changes between charging and discharging modes are independently applied on each cell when the previous cycle finished.The process finishes when the programmed number of cycles were executed.

### 2.1. Architecture

The architecture of this IoT platform consists of a series of components or layers that form the battery charge and discharge autonomous system, as illustrated in Figure 1. This layering of the platform provides scalability and flexibility to the system. The component closest to each of the battery cells is the hardware layer. The motherboard involves two symmetrical boards mounted on a single PCB to reduce manufacturing costs. The system is fully scalable by adding one more board for each pair of additional cells included in the battery. This hardware layer is responsible for acting directly on the battery and collecting all the information. In addition, it hosts the logic of the charging and discharging processes in an integrated microcontroller chip. In addition to managing the charging and discharging processes, the microcontroller is programmed to monitor and collect parameters such as voltage, intensity, and temperature, providing information about (i) the state of the process, (ii) the state of the cell, and (iii) the recovery capacity of each cell. For the implementation of this hardware layer, an Arduino ecosystem is used, giving a series of tools that facilitate development in embedded systems.

The hardware layer mainly communicates with the application layer or back-end layer, by sending and receiving messages. This application layer is responsible for hosting all the logic of the platform and its storage. To carry out this flow of messages, the communication protocol MQTT [28] is selected. This protocol is one of the most used in the field of IoT, due to its minimal packet overhead and its easy implementation. It is a messaging transport protocol based on the publisher/subscriber model. This model allows decoupling the publisher from the subscriber, avoiding any direct contact between them. In MQTT, the connection between the publisher and subscriber is handled by the broker. This broker is in charge of filtering all incoming messages and distributing them to their respective subscribers. In this publisher and subscriber model, both the hardware layer and the back-end layer act as a publisher and subscriber, due to the established two-way communication. This MQTT broker is thus part of the general architecture of the system, being located between the hardware layer and the back-end layer and allowing their bidirectional communication. For its implementation, the free code broker called Eclipse Mosquitto is used. The message serialization format used in this communication is JSON, which is a data exchange format that defines a readable and simple syntax with wide use in the field of the Internet of things (IoT) [29,30]. Recent examples to provide a reliable IoT network with efficient data transmission based on a wireless sensor network (WSN) can be found in [31], where the proposed solution combined the Artificial Bee Colony (ABC) and Gravitational Search Algorithm (GSA) to accomplish the efficient cluster head selection.

The application layer or back-end layer is responsible for hosting the platform’s logic and offering registry and storage services. It is hosted on a personal computer connected via a USB port to the communications port of the Arduino board in the hardware layer. It offers a REST API that allows the user interface layer to interact with the logical system of the platform. The REST API is an architecture style for communication among distributed systems, promoting scalability. It is used by many Internet services, using the HTTP (Hypertext Transfer Protocol) to obtain data or to execute operations.

In our solution, this interface is used by the presentation layer or front-end to operate on each of the entities stored in the database, thus offering the user access to each of them. For real-time monitoring of the process, SSE (server-sent events) technology is used, which allows the client to receive automatic updates from the server. It is used for the development and implementation of the back-end layer [32]. It is based on Java and makes use of the Spring Boot module, which facilitates the development of a web application. For the implementation of the MQTT agent and the SSE technology, the Maven dependency is integrated into the Java Project.

The presentation layer or front-end is the layer closest to the user. It offers the graphical interface with which the user interacts while using the upload and download management platform. It presents the views related to the consultation and creation of processes, configuration, real-time monitoring, etc. The graphical interface of the battery charge and discharge management platform presents a dashboard with a variety of panels that offer a variety of functionalities. These features include process creation, generation of a list of all processes, showing specific processes with graphics included, platform configuration, and real-time monitoring. The implementation of the graphical interface is carried out with Angular technology. This platform offers a versatile solution when it comes to adding cells and allows the parameterization of loading and unloading in this IoT solution. It must be taken into account that the periodic observations in each of the cells must be stored and, therefore, must be managed efficiently since they involve a large exchange of information in the communication channels, avoiding congestion.

### 2.2. Hardware

As was introduced in Section 2.1, this hardware layer is the closest layer to each cell of a multimodule battery. The design of this layer includes the selection of components to host the logic of the charging and discharging management processes, as well as to monitor and act on each battery cell. The hardware layer is thus a physical layer implemented on two types of PCBs: (i) a PCB designed by the authors where the electronic circuits that govern the charging and discharging processes of each individual cell (control and protection) are integrated, and also hosts the measurement devices (data collection); and (ii) a PCB implemented on an Arduino board (management, monitorization), including as many additional boards as were necessary according to the total number of cells to be managed by each specific battery. The processes are carried out by means of a series of sensors and actuators connected to the electronic boards, providing bidirectional communication in which the microcontroller receives information from the sensors while sending commands to the actuators. In this section, the functionalities of each component of this hardware layer are detailed, i.e., those based on the microcontroller and each of the connected electronic devices. The electronic board is an Arduino MEGA [33], with a relevant number of digital pins. The Arduino is connected to a series of sensors and actuators, as was introduced in Section 2.1. Regarding the sensors, the INA219 device manufactured by Texas Instruments is used, which monitors the voltage drop in the shunt resistor and the bus supply voltage, current, and power. This sensor performs the calculation using calibration values and multiplicative registers. It is used to measure the voltage drop between the terminals of each individual cell, monitoring the corresponding charging/discharging current. This device avoids the need to use different multimeters in each cathode and anode of the cells, since it is a board assembled with all the components pre-soldered [34]. Figure 2 shows these selected sensors. One of the main functions of the proposed system is to carry out the charging and discharging processes of the cells under controlled constant intensity. It is achieved according to the information received by the INA219, and by controlling the base intensity of a transistor that controls the charging or discharging current—circulating between the emitter and collector of the transistor. An NTC thermistor is selected to measure the temperature of each individual cell, due to its low cost, performing the calibration between the values of 20 °C and 78 °C and adjusting it by means of a fourth-degree polynomial. To transmit the information from the sensors, a digital-to-analog converter (DAC) is used, specifically the MCP4725 DAC [35] as can be seen in Figure 3.

For communication between the electronic board and these sensors, the I2C (inter-integrated circuit) bus is used. Each sensor has an I2C address that identifies it on a shared bus, since they are compatible with this communication protocol. It is a communication bus found on most Arduino boards [36], which allows communication between different devices that support this technology. This device has two pins related to the I2C address. In an I2C bus, each device must have a different address to carry out communication between devices in the same shared I2C. Pins A0 and A1 allow up to two bits of these addresses to be modified. Therefore, the maximum number of INA219 devices to be used on the same I2C bus is four. Considering that the system prototype must be scalable up to 28 cells, the problem of conflict among different addresses is solved in order to be able to share 28 INA219 devices in the same I2C through the integration of a multiplexer.

Regarding the configuration of the I2C address of an INA219, it is necessary to solder the jumpers of pins A0 and A1; by default, the device acquires the address 0 × 40, i.e., without soldering or modifying the board, depending on the configuration of the solder that is made between the pins, the device acquires one address or another. The Arduino board has digital input and output pins, as well as analog input pins, but not output pins. This results in the Arduino board being unable to generate analog signals. Arduino tries to emulate analog signals using PWM modulation, but it can only provide a digital output of −V_cc_ or V_cc_, i.e., 0 V and 5 V. This emulation can be enough in a wide variety of applications, but not in this specific case, since it is necessary to obtain a real analog signal. Due to the need to generate real analog signals, the MCP4725 device is used, which is a 12-bit DAC that gives an analog signal from the Arduino controlled by the I2C communication channel. It features non-volatile memory (EEPROM) that can be used for configuration data via the I2C channel. It also allows the last entry code entered to be stored to maintain the voltage level after any power failure. The purpose of this DAC converter is to control the charging or discharging electric current of each independent cell. Lastly, it behaves as an actuator on the specific cell, regulating its electric current in each charging or discharging cycle. The MCP4725 device address is made up of a 7-bit structure. The first four bits are the so-called device-bits, and they are not configurable (1100 for this specific device). The next two bits (A2 and A1) are defined in the manufacturing process (not modifiable afterward). The seventh bit, called A0, is configurable through pin A0. It can be connected to the VCC or VSS to modify its value between 0 and 1. A maximum of two MCP4725 devices can be simultaneously used on the same shared I2C bus. Therefore, both the INA219 and the MCP4725 devices take on the role of slaves and dedicate themselves to listening and carrying out the orders coming from the master node. I2C is based on two signal lines: a dedicated clock line (SCL) and a dedicated data line (SDA). Devices connected to the I2C bus have a unique address that identifies them. As mentioned in previous sections, the limitation when configuring the addresses of each of the devices requires the use of a multiplexer to scale the prototype and achieve a final product made up of 28 modules: 1 for each cell.

### 2.3. Software

The developed software is in charge of carrying out the battery management processes. They are mainly based on managing a series of consecutive charging and discharging cycles in each individual cell, evaluating different parameters: voltage, intensity, and temperature; and monitoring the state of the process, the state of the cell, and the recovery capacity and performance of such cells. Each charge (or discharge) cycle is defined by a charge (or discharge) curve that defines the charge intensity according to the current cell voltage. This process provides a wide variety of charging and discharging schemes that are very useful for the study of the best charging or discharging curve with the aim of finding the optimal configuration that gives the greatest improvement performance for a specific cell. To stop charging and discharging cycles, there are some values that can be modified. In our particular case, as they are focused on NIMH cells with a nominal voltage of 7.2 V, it is established that the cell is fully charged when reaching 8.1 V [37] and fully discharged when reaching 3.2 V [38]. Safety parameters are also established, such as cell temperature. In this case, a maximum temperature of 42 °C was established; the process would stop if this temperature was reached. The flowcharts of cell and system control can be seen on Figure 4 and Figure 5, respectively.

The development environment used to carry out the programming of the microcontroller is called Arduino IDE3. This application is written in the JAVA language and is used to write and upload the programs to Arduino-compatible boards. This IDE supports the C and C++ programming languages. Programs written in this programming environment are called sketches. They are stored with the ‘.ino’ extension. Programs can be developed for all Arduino boards as long as the model and corresponding serial port are indicated. In addition, it is possible to add extra libraries that supplement the functionality of the program, thus allowing data to be manipulated, as well as communications to be established, along with other functionalities. The sketch is uploaded to the microcontroller of the Arduino board through the serial port. This port can be used for board communication with any device that accepts this type of communication. The programming environment provides a serial monitor that allows interaction with the board in real time by sending a string of characters.

The flow of messages shared between the back-end layer and the Arduino is serialized using a format that allows rapid processing of these and whose readability is simple and apparent. The selected format is JSON, which is a data exchange format that defines a syntax for serializing objects, arrays, numbers, strings, Booleans, and nulls. It is based on the syntax of JavaScript, although it has some differences. It is the most used format in the IoT (Internet of things) due to its light weight when it comes to storing and transporting information [39]. In addition, its basic syntax implies an easy interpretation for both humans and machines. These properties make JSON the ideal format for exchanging messages between Arduino and the back-end layer. ArduinoJSON is an open-source library with an MIT license that allows processing JSON documents on an Arduino-based board. This library provides all the necessary tools for the serialization and deserialization of documents with JSON format. It also allows optimizing RAM memory consumption in the embedded system. It is compatible with any C++ code, being independent on any auxiliary library or Arduino. It can be used in any embedded system with C++ code. The Arduino-compatible PubSubClient library is used as well. It provides an MQTT client that allows the publication and subscription of MQTT messages on a server that supports this protocol. Initially, this library was used to communicate the Arduino and the back-end layer through the MQTT broker, based on an Arduino board with an MQTT client. To carry out this integration of an MQTT client on the Arduino board, an auxiliary board (Arduino Ethernet Shield) was included in the prototype. This additional board was integrated into the main Arduino MEGA board, providing the possibility of connecting to the Internet in a few steps through an RJ45 interface.

## 3. Case Study

A preliminary prototype was developed with four integrated circuits to evaluate the operation of the proposed system in terms of the measurement of voltage, intensity, temperature, and energy parameters; as well as the capacity of the system to withstand the operating conditions during the necessary cycles. In addition, the proposed solution is able to automate both charging and discharging cycles. As can be seen in Figure 6, each designed system is capable of operating in an autonomous and automatic manner on two battery cells in a totally independent way. The solution can then operate on eight cells of a battery. As was discussed in Section 2, the selection of mounting two circuits in one board was based on economic criteria, when manufacturing integrated circuits. This system is fully scalable, with the aim of creating a system capable of governing the charge and discharge cycles of a battery containing any number of cells, operating on each cell independently in a subsequent phase, simply by adding integrated circuits. The circuits were mounted on a rack, as shown in Figure 7 and Figure 8. The Arduino board that governs the integrated circuits was mounted on the same rack. The performance tests were carried out on several cells of a Toyota Prius battery (Figure 9); these NiMH cells have a nominal voltage of 7.2 V and are internally made up of six 1.2 V batteries.

The tests consisted of programming consecutive charge and discharge cycles with different cells in the system, establishing various charge voltage limit values, and at different charge and discharge electric current values, monitoring the temperature of the cells of the battery and the corresponding charge and discharge curves obtained on each individual cell. It was also verified that the data provided to the system by the user in the interface were transmitted to the hardware, being possible to automate uploads and downloads. Figure 10 gives a general overview of the tested EV commercial battery and the proposed system prototype. In this case, a Toyota Prius Battery made up of a total of 28 cells was selected. The shape of each of these cells and the prototype connections can be seen in Figure 11. Each pair of battery cells is governed by one of the integrated circuits shown in Figure 6 and previously described.

## 4. Results

Firstly, the obtained results were for evaluating the system performance in terms of the functions entrusted to it: carrying out the programmed automatic charging and discharging cycles at a specific constant electric current. It was also verified that the system is capable of carrying out such charging and discharging cycles with different constant electric current values, based on the assigned voltage thresholds. Subsequently, the system can be programmed to carry out charge or discharge cycles at a certain constant intensity, and, upon reaching a certain voltage value, it automatically changes to another intensity to continue the charging or discharging process until reaching the next voltage threshold or the corresponding conditions to finish the test. Therefore, this system is capable of characterizing an individual cell with NIMH technology and detecting potential cell performance failures of such individual components from a global battery. Secondly, the results obtained were observed when the integrated circuits were operating under extreme conditions, aimed to determine their ability to work under such operating conditions. Accordingly, it was tested if they were able to carry out charges and discharges at an electric intensity of 1 A, measuring the temperatures reached on the plates by using thermography. Additionally, it was evaluated if the designed integrated circuits were able to take into account the heat sinks of the resistors and transistors. In Figure 12, the temperature reached before the installation of heat sinks on the plate can be seen in the thermograph. Other results were focused on the measurement curves of voltage, intensity, temperature, and energy during charging and discharging cycles at different values of electric intensity and with different voltage thresholds (Figure 13, Figure 14, Figure 15, Figure 16 and Figure 17).

More specifically, and as can be seen in Figure 13, the discharge cycle of an individual cell was evaluated at a constant intensity of 500 mA. According to the results, the temperature during the discharging cycle was constant, deducing normal performance conditions of the cell. The energy discharged in this period is represented as well, reaching values close to 14 Wh at the end of the total discharge cycle. In Figure 14, the charge cycle of an individual cell is represented, at a constant electric intensity of 500 mA. The temperature during such charging period was constant, providing normal performance conditions of the cell. The energy used to charge the cell is also included as an additional result, reaching values close to 18 Wh at the end of the total charge cycle. Similar results can be found in Figure 15, in this case, with a constant electric intensity of 600 mA.

Figure 16 shows an example of a discharging cycle test of an individual cell, under constant electric current but including different steps: 500 mA, 300 mA, and 200 mA. The temperature during this discharging cycle was constant, and the energy discharged reached values close to 18 Wh at the end of the total discharging cycle. This test shows the suitability of the proposed system to evaluate the individual cell performance at different electric current values. An additional example of these tests can be found in Figure 17, considering other electric currents: 600 mA, 400 mA, and 300 mA. From these results, the potential consequences of carrying out an overcharging process were evaluated, at a high intensity of 600 mA and at a threshold voltage much higher than the recommended 10 V. The proposed system could thus withstand such operating conditions, though the temperature of the cell increased once fully charged (Figure 18) and possible physical deformation of the cell structure occurred, causing its structure to bulge (Figure 19).

## 5. Discussion

This proposed solution is based on the IoT technology application capable of automating the successive charging and discharging of each cell of a battery and diagnosing the individual cell status in order to proceed with possible regeneration. A complete cell regeneration process can take about 3 complete discharges and the corresponding charging processes, with an average of 4 h for each cycle. The global process can last 24 h, which is not cost-effective if not automated, since the cost of an operator for 24 h does not justify the possible regeneration of a single battery cell. It was observed that the Internet of Things is an important enabler of projects in this field as it enables conducting projects of all kinds. In addition, the versatility and ease offered by the Arduino ecosystem were demonstrated, attracting less experienced users. It was also verified that the devices (sensors and actuators) were chosen correctly, as they could obtain sufficient precision to automatically perform these loading and unloading tasks. From the proposed and evaluated system, the following step is to scale the proposed solution to a minimum of 28 cells, i.e., 14 integrated circuits governed by an Arduino board, whereby the system was able to simulate the real EV battery of a commercial solution. Initially, the same case study could be selected: Toyota Prius, which was the first hybrid car commercially available. Accordingly, it is the hybrid car with a relevant presence on the current market, and thus with the highest potential number of batteries to diagnose and regenerate. Nowadays, the assembly and prototype depicted in Figure 10 has been tested, but it is desirable to carry out the diagnosis and regeneration process of the eight battery cells simultaneously and decrease the time costs considerably, which is currently a field of interest for the authors. Another remarkable improvement for future versions of the prototype would be focused on the integration of battery diagnostic estimation algorithms, such as the State of Charge (SOC) or State of Health (SOH), by following recent contributions [40].

## 6. Conclusions

This paper describes and evaluates a low-cost system to characterize individual cells of commercial electric vehicle batteries by identifying abnormally performing cells that are out of use, thus minimizing regeneration costs in a more sustainable manner. A platform based on the IoT technology is developed, allowing the automation of charging and discharging cycles of each independent cell according to some parameters given by the user, and monitoring the real-time data of such battery cells. The proposed system performs a discharge to each individual cell, in a controlled and automated manner, by using a constant electric current until reaching a minimum cell voltage value. A preliminary prototype was developed with four integrated circuits to evaluate the operation of the proposed system in terms of the measurement of voltage, intensity, temperature, and energy parameters, as well as the capacity of the system to withstand the operating conditions during the necessary cycles. A hardware layer, application layer, and presentation layer have been developed by using low-cost hardware components and well-known protocols, such as Arduino MEGA, HTTP, MQTT agent, or I2C bus. The system allows for substantially reducing this process, automating and minimizing human resources, and providing a cost-effective and sustainable manner to regenerate individual cells and extend the use of EV batteries. A case study based on a commercial EV Toyota Prius NiMH battery is included in the paper. Numerous discharging cycles are given, under controlled parameters (constant electric current, variable-step electric current). All electric variables are also monitored and stored, as well as temperature and energy. The proposed system allows for carrying out tests under over-charging conditions, to stress the cells and evaluate their performance under such abnormal circumstances. The proposed system is able to be used with other commercial EV batteries and provide a friendly interface with the user.

For future contributions of the proposed solution, the authors are currently working on the following items:The integration of battery diagnostic estimation algorithms, including further analysis of the monitored data.The evaluation with other commercial EV batteries and different charging and discharging conditions.The extension of IoT technology toward a variety of communication protocols and devices.

Data and scripts are available to the scientific community for any research purpose.

## Figures and Tables

**Figure 1 sensors-23-00816-f001:**
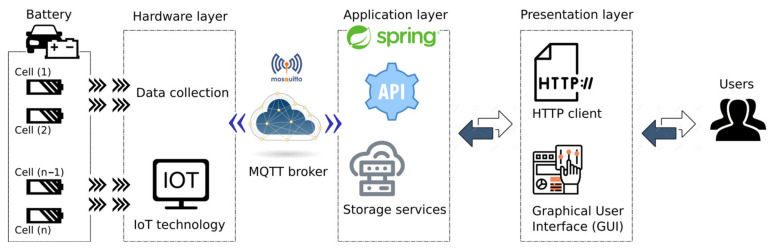
System architecture and its components.

**Figure 2 sensors-23-00816-f002:**
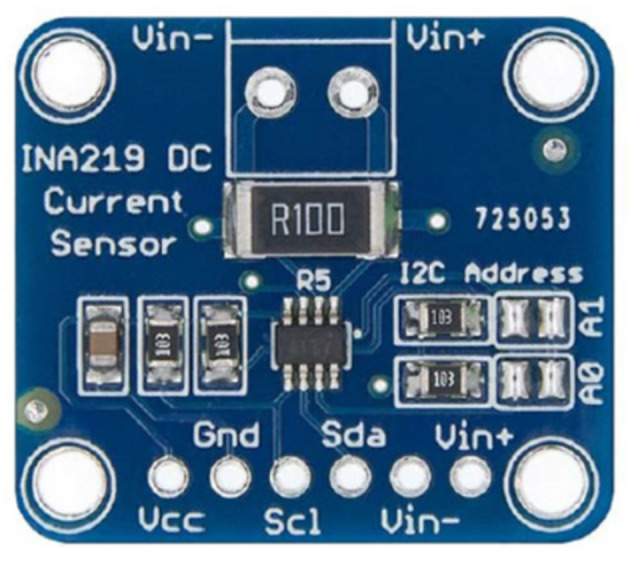
INA219. Source: [34].

**Figure 3 sensors-23-00816-f003:**
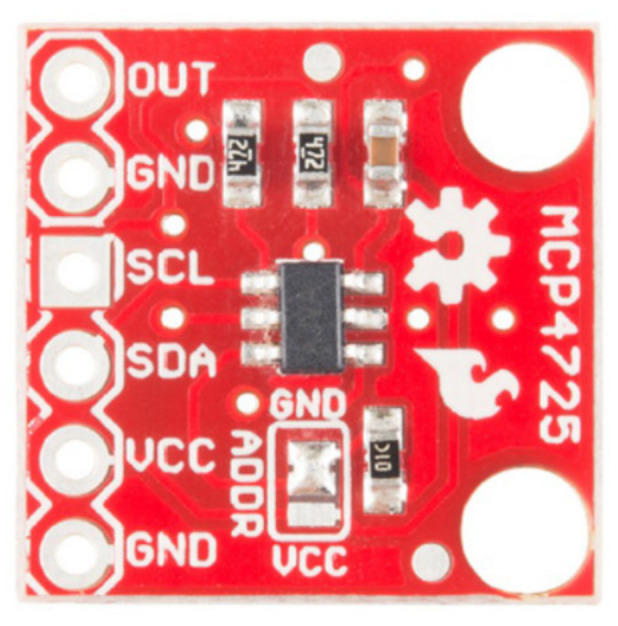
MCP4725. Source: [35].

**Figure 4 sensors-23-00816-f004:**
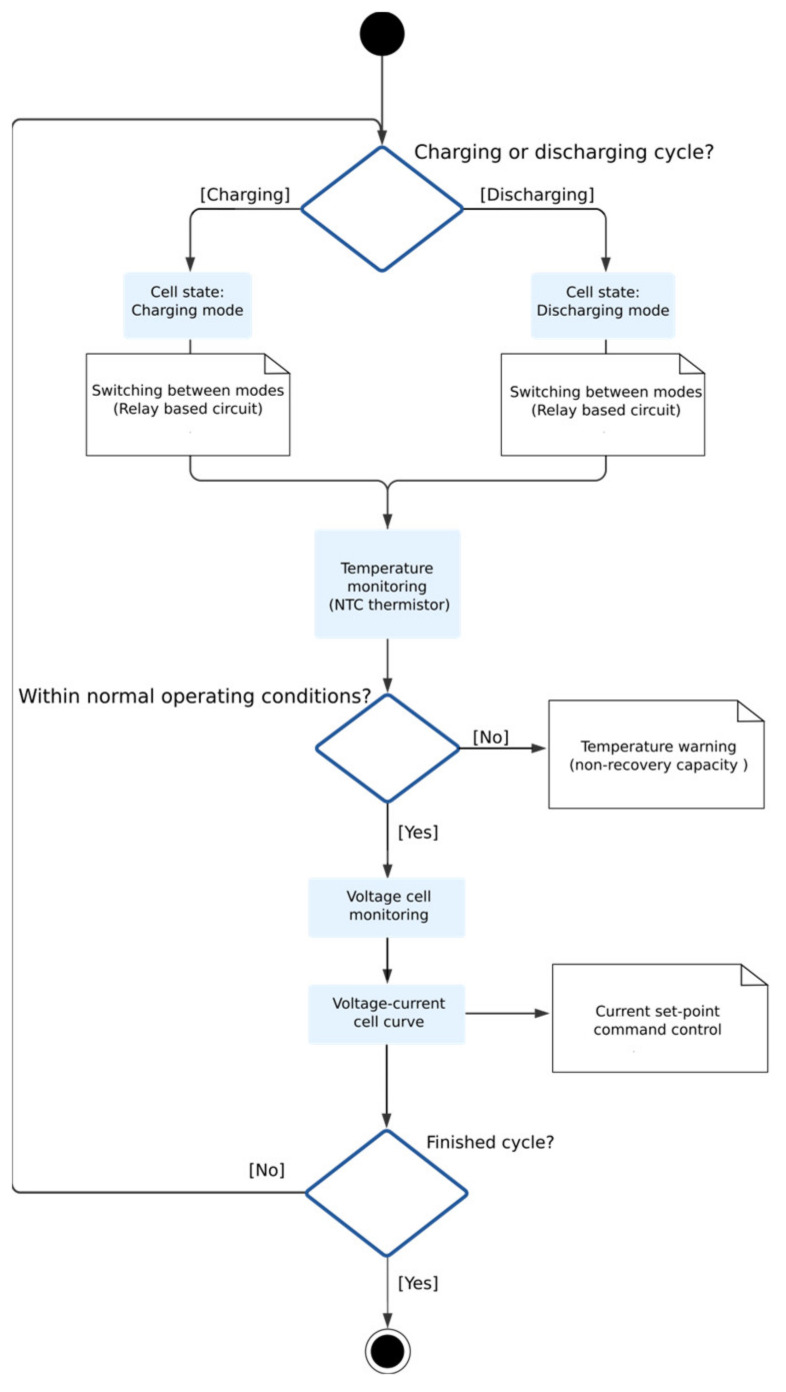
Cell control flow chart.

**Figure 5 sensors-23-00816-f005:**
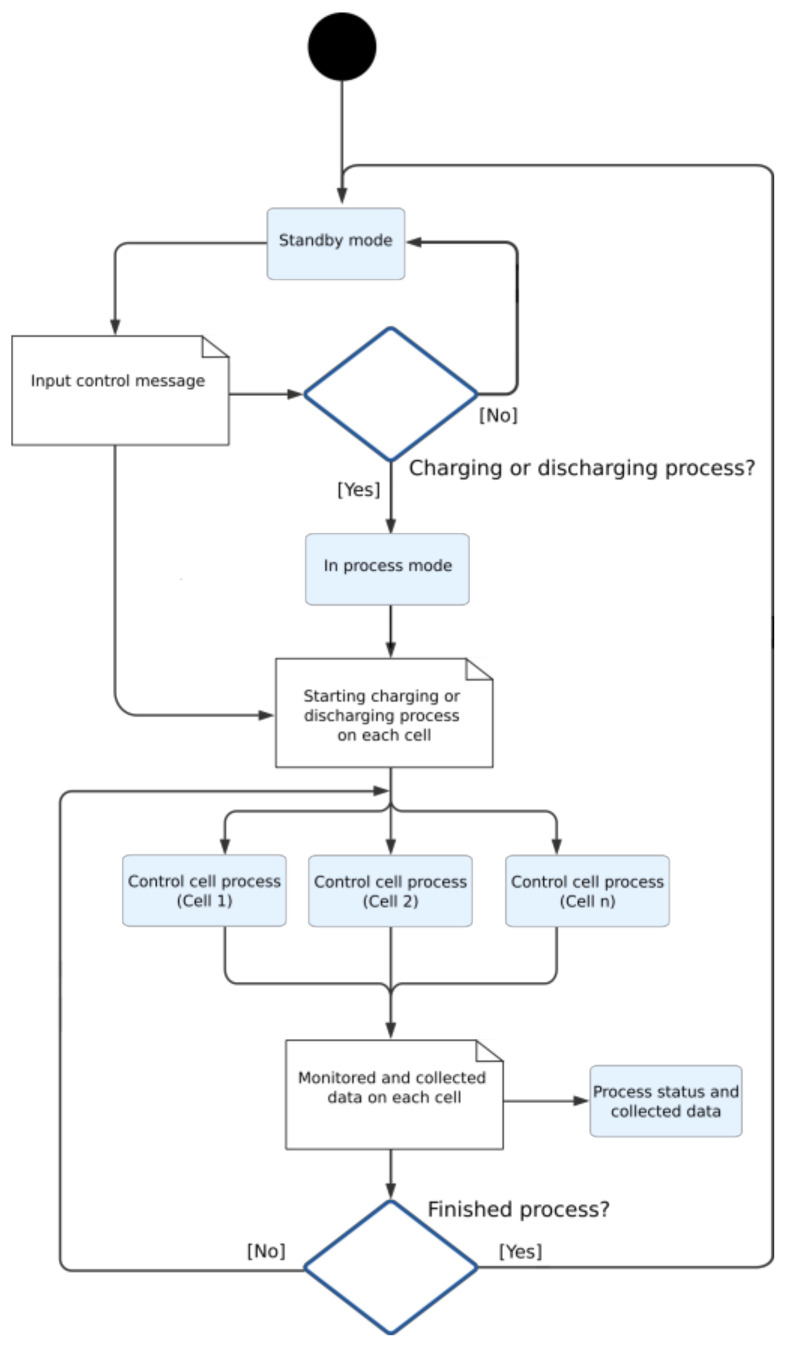
System control flow chart.

**Figure 6 sensors-23-00816-f006:**
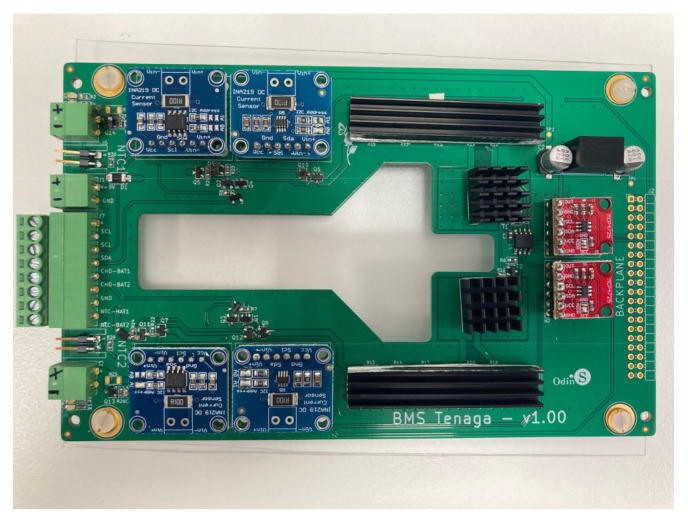
Designed PCB to control the charging and discharging cycles of each cell.

**Figure 7 sensors-23-00816-f007:**
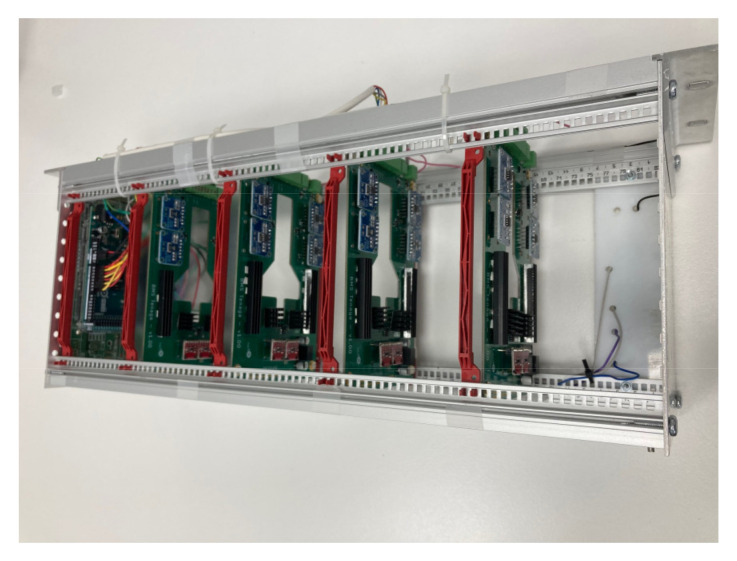
Integrated circuit assembly in a rack with the Arduino card. General overview.

**Figure 8 sensors-23-00816-f008:**
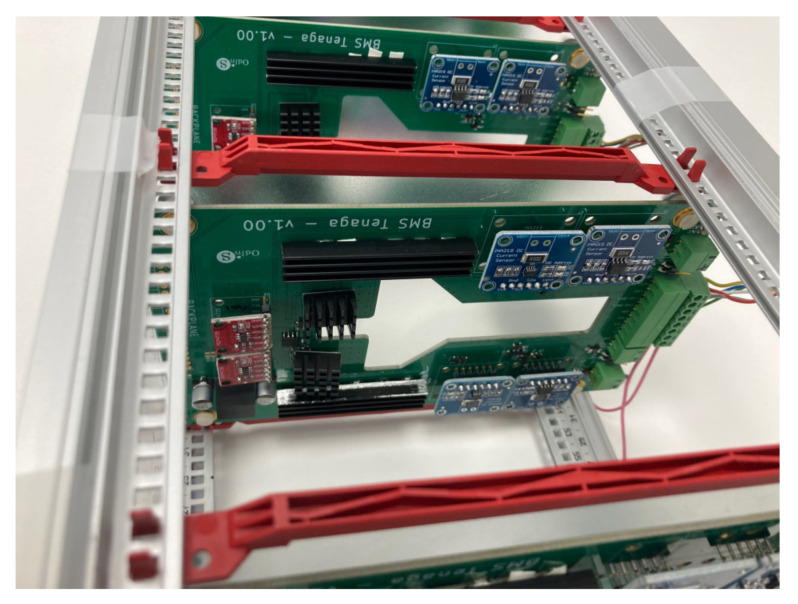
Detailed view of the assembly of a designed PCB mounted in the rack.

**Figure 9 sensors-23-00816-f009:**
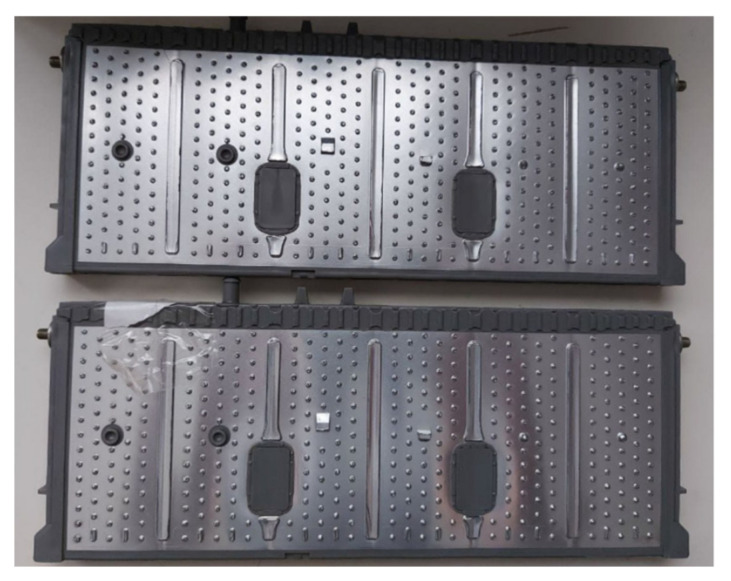
Details of two commercial cells tested in this case study: Toyota Prius battery cells.

**Figure 10 sensors-23-00816-f010:**
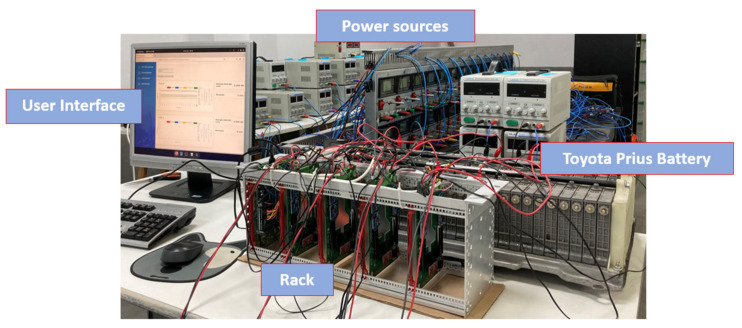
Toyota Prius battery and proposed system prototype. General overview.

**Figure 11 sensors-23-00816-f011:**
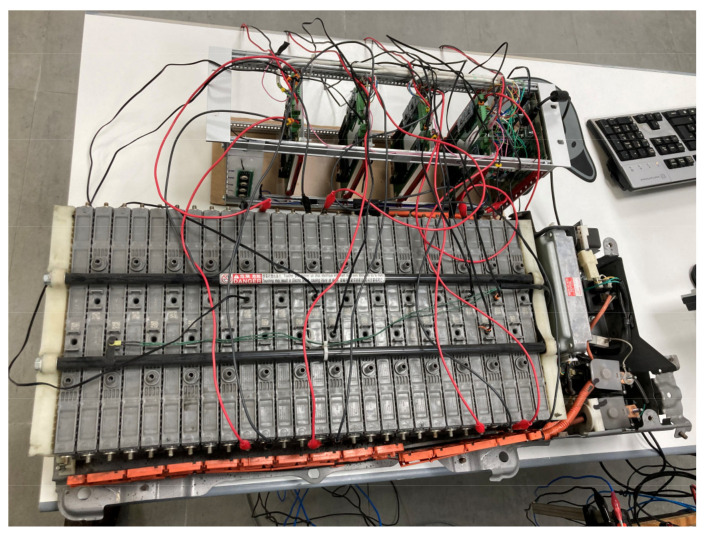
Toyota Prius battery and prototype connections to the battery cells.

**Figure 12 sensors-23-00816-f012:**
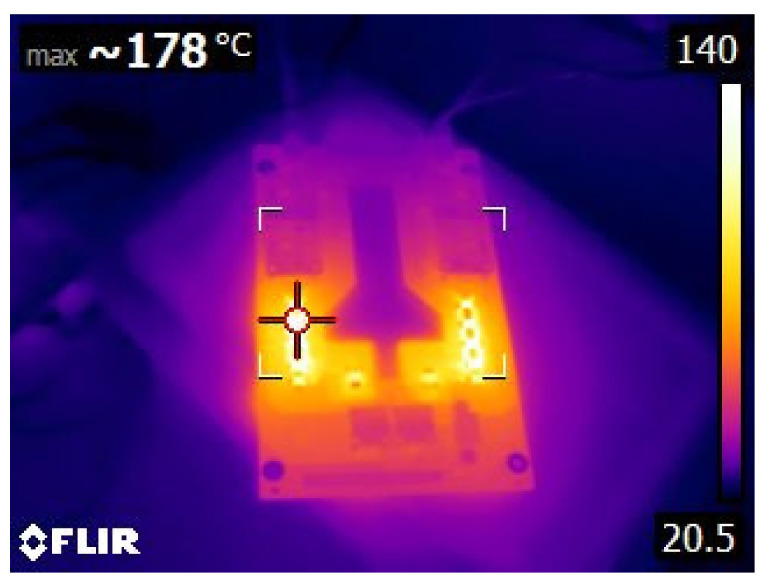
Detailed thermograph of an integrated circuit before the installation of the heat sinks.

**Figure 13 sensors-23-00816-f013:**
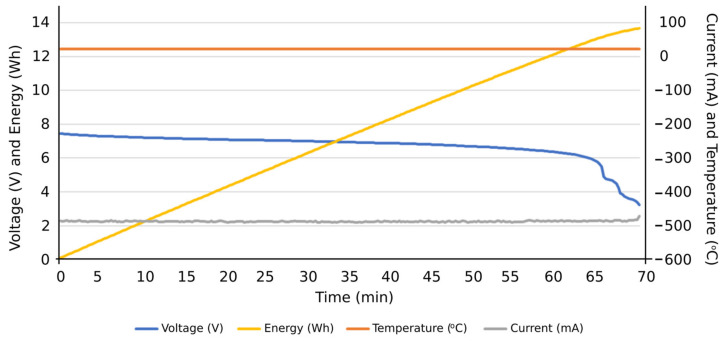
Results of a discharging cycle at constant intensity of 500 mA. Individual cell.

**Figure 14 sensors-23-00816-f014:**
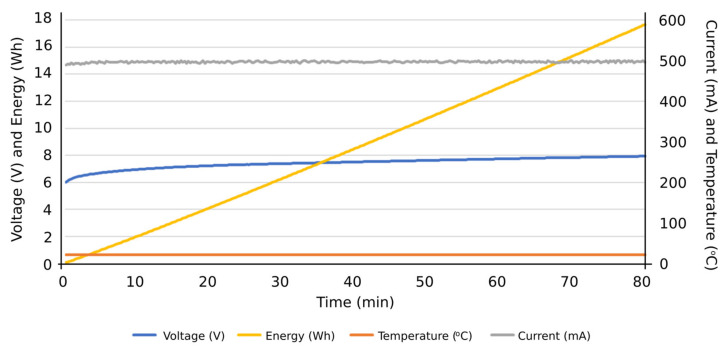
Results of a charging cycle at constant intensity of 500 mA. Individual cell with a threshold voltage of 8 V.

**Figure 15 sensors-23-00816-f015:**
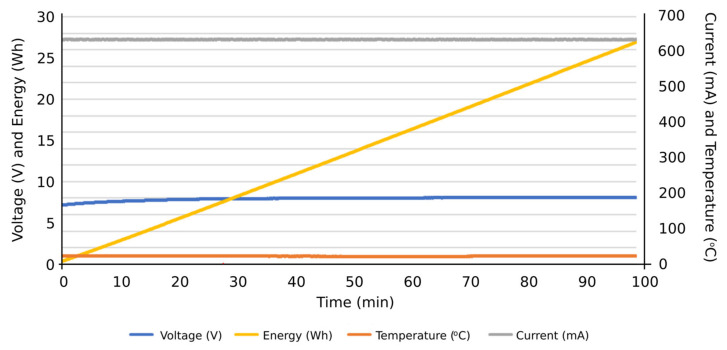
Results of a charging cycle at a constant intensity of 600 mA. Individual cell with a threshold voltage of 8.1 V.

**Figure 16 sensors-23-00816-f016:**
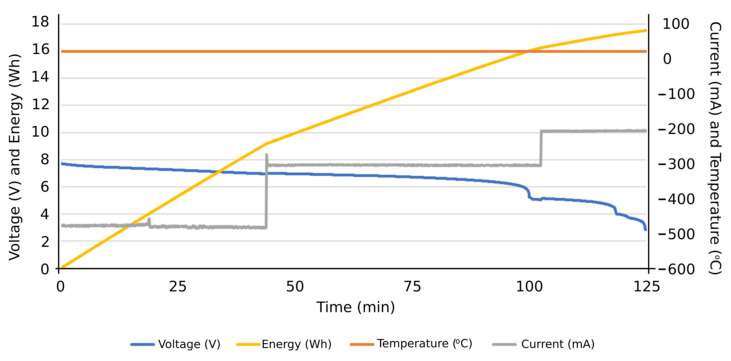
Results of a discharging cycle at different constant intensities: 500 mA, 300 mA, and 200 mA.

**Figure 17 sensors-23-00816-f017:**
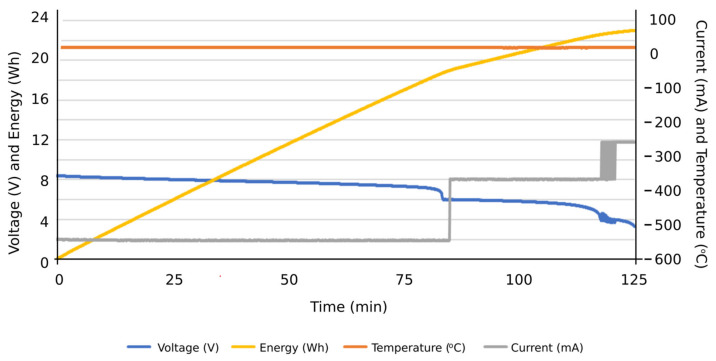
Results of a discharging cycle at different constant intensities: 600 mA, 400 mA, and 300 mA.

**Figure 18 sensors-23-00816-f018:**
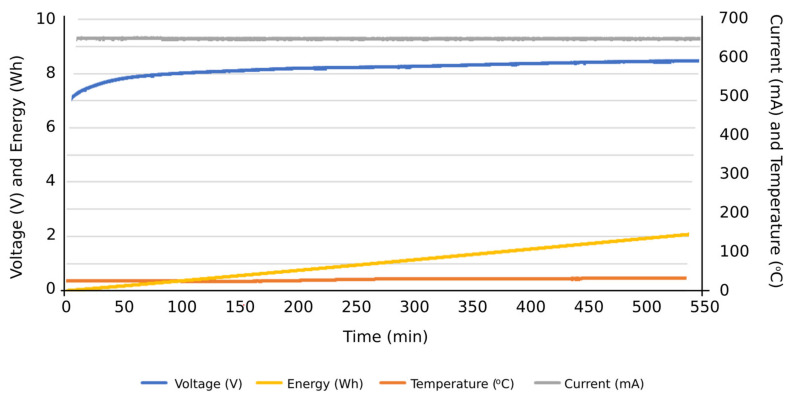
Results of a charging cycle at a constant intensity of 600 mA with a threshold voltage of 10 V.

**Figure 19 sensors-23-00816-f019:**
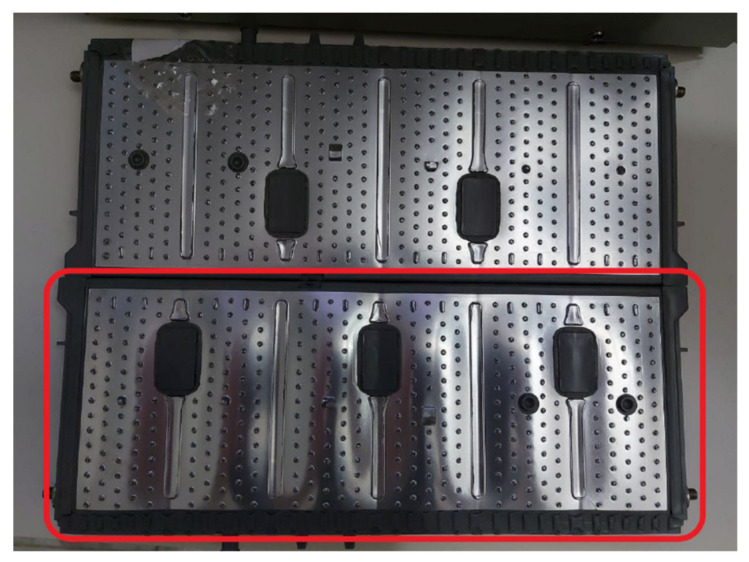
Comparison of two cells; the one below was deformed by an over-charge process.

## Data Availability

The data presented in this study are available on request from the corresponding author.

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
