# Peer review of "A Low-Cost Hardware Architecture for EV Battery Cell Characterization Using an IoT-Based Platform"

_sensors, 2023, doi:10.3390/s23020816_

Round 1

Reviewer 1 Report

A Low-Cost Hardware Architecture for EV Battery Characterization Using an IoT-Based Platform

The authors should explain more clear about identifying potentially abnormal cell performance and providing an alternative way to recover batteries.

Figure 1. System architecture and its components. Should be more clear and informative

In MQTT, the connection between the publisher and subscriber is handled by a third party what is the third party considered ?

The third result obtained was the measurement curves of voltage, intensity, temperature, and energy during charging and discharging at different values of charging and discharging intensity and with different voltage thresholds. The authors should mention the units

HTTP (Hypertext Transfer Protocol) protocol to obtain what kind of data and execute what operations.

Authors can refer

Energy efficient cluster head selection for internet of things

Author Response

Please, find attached our response. Thank you.

Reviewer 2 Report

In my opinion, manuscript is in very poor condition. The study has many serious technical issues that need to be addressed and might be marginally be accepted by editors only if required amendments can be done by authors in final submission. In that case, the article may not be published in its current shape because the contents and results does not meet the requirements for Sensor journal publication.  

 1.      Overall, the abstract is poorly written, and the major contributions and identified problems are not highlighted.

2.      The authors must revise their paper's introduction to emphasize the originality of their work. It is unclear what is novel about this approach. On which prior works did the authors base their work? What's new? Please provide references that are more pertinent and recent, with an emphasis on similar works

3.      Line 100: “The remainder of the paper is structured…” This should begin a new paragraph.

4.      The paper is lack of mathematical part to support the proposed model.

5.      The authors should provide a more detailed explanation of the employed methodology with a modified and comprehensive diagram, which should be the primary contribution of the paper in predicting crucial results.

6.      The paper is lack of mathematical part to support the proposed model.

7.      It is unclear that what significant parameters are considered   

8.      In Figure 1, the proposed System architecture and its constituent parts look very generic and are unable to demonstrate any novel features.

9.      I don't think it's necessary to add too much explanation to section 2.2, and a table would make it much easier to explain the content than long and pointless explanations.

10.  Section 2.3; Authors again need to add some flowchart and table to demonstrate the functionality of the software part in battery management processes.

11.  All of the captions for Figures 4-6-7 need to be rewritten because they are unclear.

12.  The quality of the figures is very low; they should be added with a higher resolution for better representation and legibility.

13.  The graphs that are provided in figures 9-14 are unclear because they show the discharge occurring at a constant intensity of 500 mA in one of the cells while maintaining a constant temperature and voltage (Wh). This occurs even though the voltage is decreasing while the energy level is dramatically rising. This demonstrates that the research's values and data are based on assumptions rather than actual observations.

14.  The section on the discussion is missing and should be added as a new section before the conclusion.

The paper's conclusion should be rewritten to incorporate the key findings of the performance analysis. In its current form, it is poorly written.

1.     Authors are required to include recent references that strengthen the paper's content and are pertinent to its subject area.

Author Response

Please, find attached our detailed response. Thank you.

Round 2

Reviewer 1 Report

Paper can be accepted 

Author Response

The authors express their sincere appreciation to the Reviewer 1 for his/her positive comments and the acceptation of our contribution. Thank you.

Reviewer 2 Report

Authors responses are satisfactory. Accepted for publication.

Future recommendations are still missing at the end of Conclusion section. Please add it in bullet points.  

Author Response

The authors express their sincere appreciation to the Reviewer 2 for his/her positive comments and the potential acceptation of our work. 

Future recommendations have been included at the end of Conclusion section. Please, revise the last paragraph in blue color.

Thank you.